# Synergistic Effect in Plasmonic CuAu Alloys as Co-Catalyst on SnIn$_4$S$_8$ for Boosted Solar-Driven CO$_2$ Reduction

Zhengrui Yang [1], Jinman Yang [1], Kefen Yang [1], Xingwang Zhu [2], Kang Zhong [1], Ming Zhang [1], Haiyan Ji [1], Minqiang He [1], Huaming Li [1] and Hui Xu [1,*]

[1] School of Materials Science & Engineering, Institute of Energy Research, Jiangsu University, Zhenjiang 212013, China
[2] School of Environmental and Chemical Engineering, Yangzhou University, Yangzhou 225009, China
* Correspondence: xh@ujs.edu.cn

**Abstract:** The photoreduction of CO$_2$ to chemical fuels represents a promising technology to mitigate the current energy dilemma and global warming problems. Unfortunately, the original photocatalysts suffer from many side reactions and a poor CO$_2$ conversion efficiency. The rational combination of active co-catalyst with pristine photocatalysts for promoting the adsorption and activation of CO$_2$ is of vital importance to tackle this grand challenge. Herein, we rationally designed a SnIn$_4$S$_8$ nanosheet photocatalyst simultaneously equipped with CuAu alloys. The experimental results proved that the CuAu alloy can trap the electrons and enhance the separation and transport efficiency of the photogenerated carrier in the photocatalyst, alleviating the kinetic difficulty of the charge transfer process because of the preferable localized surface plasmon resonance (LSPR). Furthermore, the CuAu alloy works as the synergistic site to increase the CO$_2$ adsorption and activation capacity. The optimized CuAu-SnIn$_4$S$_8$ photocatalyst exhibited a superior performance with CO generation rates of 27.87 μmol g$^{-1}$ h$^{-1}$ and CH$_4$ of 7.21 μmol g$^{-1}$ h$^{-1}$, which are about 7.6 and 2.5 folds compared with SnIn$_4$S$_8$. This work highlights the critical role of alloy cocatalysts in boosting a CO$_2$ activation and an efficient CO$_2$ reduction, thus contributing to the development of more outstanding photocatalytic systems.

**Keywords:** SnIn$_4$S$_8$; CuAu alloy; synergistic effect; photocatalysis; CO$_2$ reduction

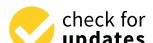



## 1. Introduction

As the scale of human activities continues to expand, global warming is known as an ever-increasingly severe global environmental problem [1]. In addition, the contradiction between the universal fast-growing energy demand and the global energy imbalance is growing acute [2]. The above problems have driven numerous researches for the sustainable transformation of CO$_2$ into hydrocarbons and highly valuable chemical products. Among the available resources, the renewable solar photocatalytic transformation of CO$_2$ via semiconducting photocatalysts shows the most promising application in simultaneously solving renewable energy production and alleviating the warming effect [3,4]. However, the photocatalytic performance of single semiconductors is hindered by a serious electron-hole pairs recombination and the poor adsorption and activation capacities of inert CO$_2$ molecules on the surface [5,6]. From this perspective, rationally developing suitable photocatalysts appears to be particularly significant.

Among the various previously reported semiconductors photocatalysts, bimetallic sulfides have aroused great scientific interest due to their unique characteristics, for instance, their adjustable morphology, enriched active sites, controllable band structure and fast photoexcited charge dynamics [7,8]. Particularly, SnIn$_4$S$_8$ is a typical n-type bimetallic sulfide semiconductor with a cubic spinel structure [9]. Featured by the narrow band gap, easily adjustable electronic and optical properties, SnIn$_4$S$_8$ is a suitable candidate in high-energy batteries and photocatalytic applications [10,11]. The above advantages

inspire us to consider their potential roles in photocatalytic $CO_2$ reduction. However, the poor light utilization and short lifetime of photoinduced carriers seriously influence the photocatalytic ability in visible light. Therefore, $SnIn_4S_8$ needs to be modified to suppress fast recombination of photoinduced carriers and enhance the ability to absorb and activate $CO_2$ molecules [12].

Decorating cocatalysts on the photocatalyst surface represents one the most appealing means to enhance the $CO_2$ photoreduction performance. On the one hand, cocatalysts can serve as electron sinks to capture photoexcited electrons, which accelerates the efficient charge separation and simultaneously restrains the occurrence of side reactions [13,14]. On the other hand, cocatalysts can act as extra active sites to lower the activation potentials of $CO_2$ molecules for the enhancement of the adsorption and activation of reactants, thus making for the enhanced $CO_2$ conversion efficiency [15,16]. Up to now, various metal cocatalysts like Pd, Au, Ag, Pt and Cu et al. have been developed to modify the semiconductor photocatalyst for a photocatalytic $CO_2$ conversion [17]. However, the function of single-metal cocatalysts has resulted in the low selectivity of carbon-based products and sluggish enhancements in the total amounts of products [18,19]. Previous studies have demonstrated that alloys composed of two or multiple metals have huge potential for optimizing the catalytic performance via providing different local atomic arrangements on the catalyst surface to tune the adsorption configuration of the reactants and $CO_2$ and control the d-band centers of the catalyst [20–22]. The synergic effect between the various atoms of the alloy cocatalyst will offer coupled sites to stabilize the $CO_2$ molecules and accurately adjust the adsorption and desorption of crucial reaction intermediates. Moreover, the Schottky barrier that appeared at the alloy/substrate interface can realize an efficient charge separation because of the internal electric field [23,24].

On account of the high activation potential barrier of $CO_2$ molecules, as well as multiple electrons and protons transfer and side reactions involved in the process of $CO_2$ re-duction, it is anticipated that the reasonable design of alloy cocatalysts can play a synergy in terms of promoting the $CO_2$ activation and trapping the photogenerated charge [25]. Copper (Cu), widely recognized as a cocatalyst, facilitates the different forms of hydrocarbons for its strong binding capacity with $CO_2$ molecules and reaction substrates [26]. In the design of alloys, integrating Cu with noble metals (Ag, Pt and Au) is the most favorable approach, not only leading to an increase in visible light absorption but also in photoexcited electron-hole pairs separation owing to the superior localized surface plasmon resonance (LSPR) and charge storage properties [27–29]. Taken together, it is expected that the alloying of active metal Cu with plasmonic metal Au supplies an ideal platform to optimize the photoreduction reduction performance in terms of modifying $SnIn_4S_8$ [30,31].

In this work, we introduced the CuAu alloy cocatalysts supported on the $SnIn_4S_8$ nanosheet photocatalyst for an optimized photocatalytic activity in the $CO_2$ reduction to CO and $CH_4$. The introduction of the CuAu alloy's nanoparticles drives the transfer of photoexcited electrons to the co-catalyst and improves the light absorption ability. Furthermore, the AuCu alloy can act as the synergistic site to boost $CO_2$ adsorption and activation. Among the designed samples, the $Cu_{10}Au_1$-$SnIn_4S_8$ achieved the activity of 27.87 $\mu$mol $g^{-1}$ $h^{-1}$ of CO and 7.21 $\mu$mol $g^{-1}$ $h^{-1}$ of $CH_4$, which are about 7.6 and 2.5 folds compared with $SnIn_4S_8$. This work offers a new approach for designing alloy cocatalysts-modified metal sulfides for the efficient photocatalytic conversion of $CO_2$ to fuels.

## 2. Results and Discussion

The synthetic process of the $Cu_xAu_y$-$SnIn_4S_8$ is distinctly displayed in Figure 1. Firstly, the morphologies of as-synthesized $SnIn_4S_8$ and $Cu_{10}Au_1$-$SnIn_4S_8$ were investigated by scanning electron microscopy (SEM) and transmission electron microscopy (TEM). As shown in Figure 2a, it is obvious that $SnIn_4S_8$, with a wavy-like morphology, was fabricated by means of a simple hydrothermal method. Furthermore, the TEM images further demonstrate the unique structure of the $SnIn_4S_8$ nanosheet, where the multiple thin sheets with wrinkles can be clearly observed (Figure 2b). The dispersed irregular nanosheets' structure

facilitates an increase in more exposed active sites on a high surface area and shortens the charge diffusion distance to inhibit their recombination [32]. As illustrated in Figure 2c, the CuAu alloy's nanocrystals are an in-situ growth on $SnIn_4S_8$ nanosheets by the method of aqueous solution synthesis. The microstructure of $Cu_{10}Au_1$-$SnIn_4S_8$ can be further exhibited by TEM images (Figure 2d–f). The morphology of $SnIn_4S_8$ remains unaltered after the loading of the CuAu alloy. The $Cu_{10}Au_1$ alloy's nanoparticles show a round-shaped form with a uniform particle diameter of about 8 nm, which are relatively even and supported on the surface of $SnIn_4S_8$. In the case of $Cu_{10}Au_1$-$SnIn_4S_8$, a high resolution TEM (HRTEM) image shows that the characteristic spacing is 0.32 nm, which belongs to the (311) plane of $SnIn_4S_8$ (Figure 2l). The CuAu alloy nanoparticle shows the lattice distances of 0.24 and 0.21 nm, respectively, corresponding with the (201) and (211) planes for the CuAu alloys, indicating that the CuAu alloys are successfully loaded on the surface of $SnIn_4S_8$. Elemental mapping images (Figure 2g–k) and Energy-Dispersive X-Ray spectroscopy (EDX) (Figure S1) were obtained to analyze the components of $Cu_{10}Au_1$-$SnIn_4S_8$. The distribution of the S, In and Sn elements greatly corresponds to the structure of $SnIn_4S_8$. In addition, Cu and Au elements are almost distributed at the same position in the selected area, confirming the successful formation of the CuAu alloy.

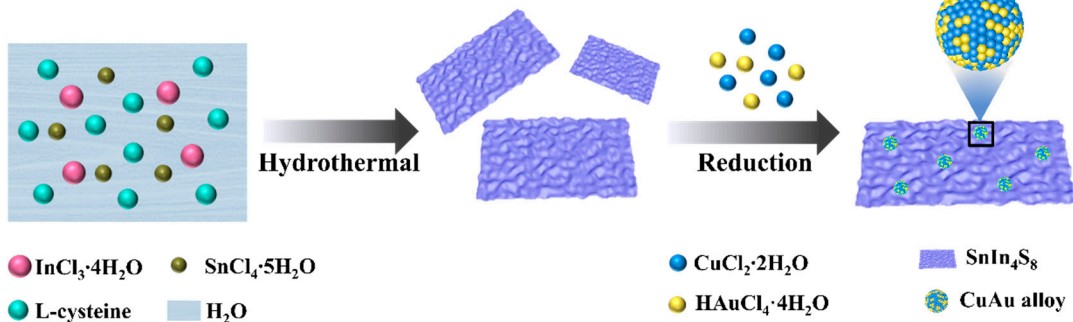

**Figure 1.** Schematic illustration for the synthetic process of $Cu_{10}Au_1$-$SnIn_4S_8$.

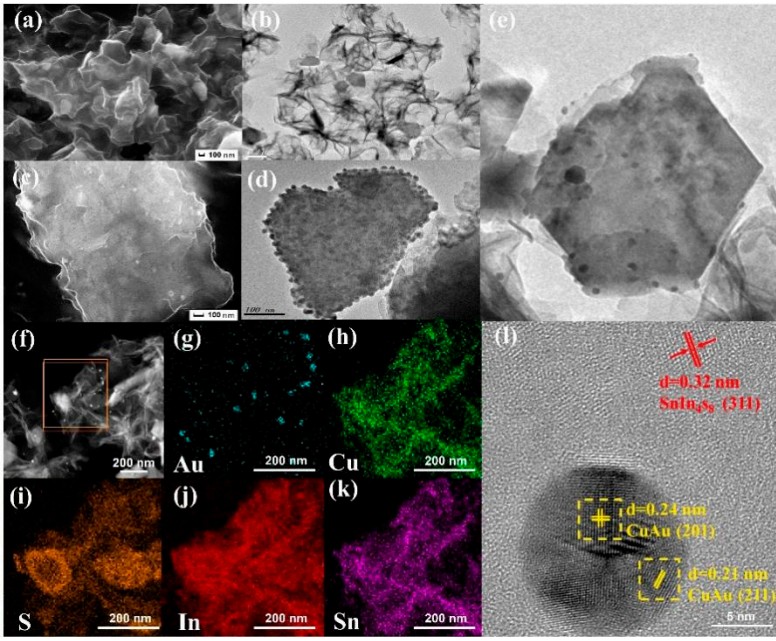

**Figure 2.** (**a**) SEM and (**b**) TEM images of $SnIn_4S_8$. (**c**) SEM and (**d–f**) TEM images of $Cu_{10}Au_1$-$SnIn_4S_8$. (**g–k**) Elemental mapping of $Cu_{10}Au_1$-$SnIn_4S_8$. (**l**) HRTEM image of $Cu_{10}Au_1$-$SnIn_4S_8$.

X-ray diffraction (XRD) was used to explore the crystal structures of the samples. As illustrated in Figure 3a, the diffraction peaks of $SnIn_4S_8$ are perfectly matched with the crystalline planes of cubic $SnIn_4S_8$ (JCPDS 42-1306), which demonstrate that $SnIn_4S_8$ was successfully synthesized. After the decoration of the Au nanoparticles on $SnIn_4S_8$, the additional peaks at 38.1°, 44.3°, 64.5° and 77.5° in the XRD pattern for $Au_2$-$SnIn_4S_8$ correspond to the (100), (200), (220) and (311) planes of Au (JCPDs 04-0784), illustrating that the Au nanoparticles were successfully supported on the surface of $SnIn_4S_8$. The diffraction peaks in the $SnIn_4S_8$ loaded with Cu at 43.2°, 50.4° and 74.1° can be indexed to (111), (200) and (220) planes of Cu (JCPDS 04-0836). Significantly, in the case of the XRD pattern of $Cu_{10}Au_1$-$SnIn_4S_8$, aside from the main typical peaks ascribing to $SnIn_4S_8$, the other characteristic peaks are well assigned to the range of the XRD patterns from Au (JCPDs 04-0784) to Cu (JCPDs 04-0836) and can be clearly observed, which present a representative characteristic for the CuAu alloy [19,26].

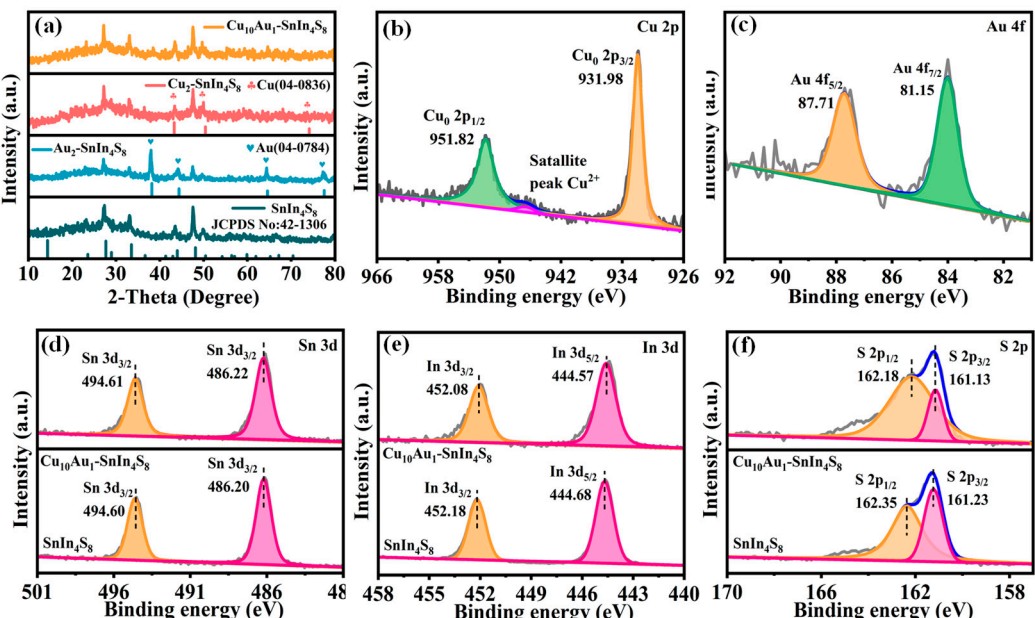

**Figure 3.** (**a**) XRD patterns of the as-prepared samples. XPS spectra for (**b**) Cu 2p and (**c**) Au 4f of $Cu_{10}Au_1$-$SnIn_4S_8$, (**d**) Sn 3d, (**e**) In 3d and (**f**) S 2p of $SnIn_4S_8$ and $Cu_{10}Au_1$-$SnIn_4S_8$.

The X-ray photoelectron spectra (XPS) further studied the chemical states of the elements and electronic states of photocatalysts. The existence of S, In and Sn, along with Cu and Au, were affirmed in the wide spectrum of both $SnIn_4S_8$ and $Cu_{10}Au_1$-$SnIn_4S_8$ (Figure S2), which agrees with accordingly elemental mapping images. As for the XPS spectra of bare $SnIn_4S_8$, there are two typical peaks in the Sn 3d spectra at about 486.01 and 494.52 eV, belonging to Sn $3d_{5/2}$ and Sn $3d_{3/2}$, respectively (Figure 3d) [9]. The In 3d spectra in Figure 3e reveal that the binding energies at around 444.40 and 452.94 eV are arisen from the In $3d_{5/2}$ and In $3d_{3/2}$ peaks [10]. The S 2p spectra clearly exhibit the two diffraction peaks of S $2p_{3/2}$ and S $2p_{3/2}$ at the binding energy of 161.0 and 162.39 eV, respectively [11]. In addition, the additional peak positioned at 163.80 eV suggested the presence of Sn (IV)-S on the surface of the samples (Figure 3f) [5]. The Au 4f levels of the $Cu_{10}Au_1$-$SnIn_4S_8$ spectra at around 81.15 and 87.71 eV correspond to the Au $4f_{7/2}$ and Au $4f_{5/2}$, which can be attributed to $Au^0$ (Figure 3c) [21]. In regard to the Cu 2p spectra (Figure 3b), the binding energy at 931.98 and 951.82 eV are ascribed to Cu $2p_{3/2}$ and Cu $2p_{1/2}$, which is in the form of $Cu^0$ [15]. In addition, a satellite peak at 946.59 eV can be indexed to Cu (II) caused by the inevitable surface oxidation of metallic Cu [26]. These XPS results further demonstrate the formation of the CuAu alloys in the $Cu_{10}Au_1$-$SnIn_4S_8$. Notably, Sn 3d, In 3d and S 2p peaks shift to a higher binding energy by different values as compared to the pure

$SnIn_4S_8$, indicating the charge transfer from $SnIn_4S_8$ to the CuAu alloy due to the difference in the Fermi levels [16]. This also suggests that there is an interface between the $SnIn_4S_8$ and AuCu alloy, which contributes to the transfer and migration of free electrons and the enhancement of the photocatalytic activity [33,34]. The phenomenon is also explored by the XPS valence band spectra (Figure S3), where the VBM of $Cu_{10}Au_1$-$SnIn_4S_8$ shifts towards a lower value compared with pure $SnIn_4S_8$, respectively, further confirming the electron transport through the interface between $SnIn_4S_8$ and the AuCu alloy [35].

In order to verify the impacts of the modified CuAu alloy, the activity of the synthesized photocatalysts in a $CO_2$ reduction including triethanolamine (TEOA) as the sacrificial agent was investigated under a 300 W Xe lamp irradiation. There were no liquid-phase products detected, and the average evolution rates of gas-phase products by diverse CuAu alloys integrated on $SnIn_4S_8$, monometallic $Au_2$-$SnIn_4S_8$ and $Cu_2$-$SnIn_4S_8$ and $SnIn_4S_8$ are compared in Figure 4a. The pristine $SnIn_4S_8$ catalyst generated relatively less CO, $CH_4$ and $H_2$, which elucidates the generation rates for 3.6 $\mu mol\ g^{-1}\ h^{-1}$, 2.5 $\mu mol\ g^{-1}\ h^{-1}$ and 0.3 $\mu mol\ g^{-1}\ h^{-1}$, respectively. With the integration of single metal nanocrystals on $SnIn_4S_8$, photocatalytic production has been significantly improved. On the one hand, the introduced metal species with an excellent electron conduction capacity can effectively enrich electrons and provide additional active sites for the accelerating $CO_2$ reduction reaction [36]. Especially, gold nanoparticles (Au NPs) can generate large amounts of hot electrons with a high energy to reduce $CO_2$ into CO and $CH_4$ owning to the strong LSPR effect, and its electron storage properties lead to enhanced electron-hole pair separation in the structure of the metal-semiconductor composition [36,37]. On the other hand, the interfaces between the metal nanocrystals and $SnIn_4S_8$ can facilitate a charge transfer to accelerate the reaction kinetics [38]. Significantly, the CuAu alloy as the cocatalyst is demonstrated to be a very beneficial catalyst which shows the obviously improved activity compared with the single metal loaded on $SnIn_4S_8$. Among the various amount of the CuAu alloy-loaded $SnIn_4S_8$, $Cu_{10}Au_1$-$SnIn_4S_8$ displays the most outstanding photocatalytic performance with CO generation rates of 27.87 $\mu mol\ g^{-1}\ h^{-1}$ and a $CH_4$ evolution rate of 7.21 $\mu mol\ g^{-1}\ h^{-1}$, which are about 7.6 and 2.5 folds compared with $SnIn_4S_8$, respectively. This result suggests that the synergistic function between the Cu and Au components in the CuAu alloys is indispensable for the photocatalytic activity improvement of CO and $CH_4$. The formation rates of each product enhance with the larger Cu/Au molar ratio because of the additional active sites. Meanwhile, the latest photocatalysts for $CO_2$ reduction and their abilities are listed in Table S1. The performance of $Cu_{10}Au_1$-$SnIn_4S_8$ is in a competitive position. However, further, the increasing Cu/Au molar ratio results in a reduced photocatalytic performance, which can be attributed to the recombination centers generated by adjoining CuAu alloys on $SnIn_4S_8$, making the immediate recombination of photoinduced carriers [39].

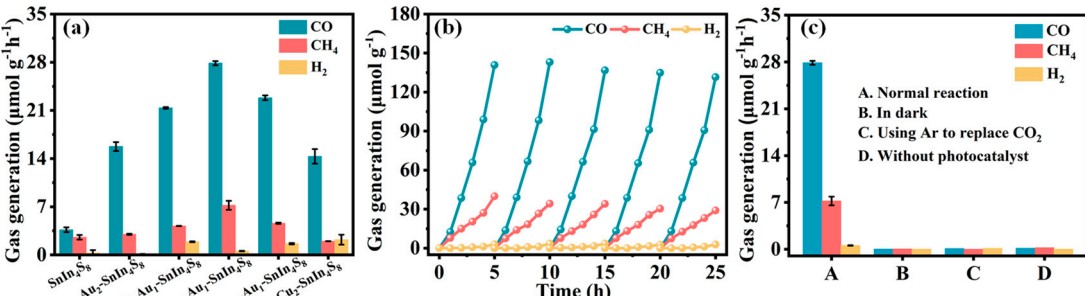

**Figure 4.** (**a**) CO, $CH_4$ and $H_2$ formation rates of the samples. (**b**) Long-time photocatalytic stability of $Cu_{10}Au_1$-$SnIn_4S_8$ for 25 h. (**c**) Comparison of photocatalytic $CO_2$ reduction performance in different conditions.

Apart from the excellent photocatalytic activity, the $Cu_{10}Au_1$-$SnIn_4S_8$ also exhibits outstanding photocatalytic stability. As presented in Figure 4b, $Cu_{10}Au_1$-$SnIn_4S_8$ still holds

great durability after 25 h with 5 recycling tests, illustrating that $Cu_{10}Au_1$-$SnIn_4S_8$ catalysts are suitable for a photocatalytic $CO_2$ reduction because of the well-maintained structure and shape. The XRD and SEM patterns of $Cu_{10}Au_1$-$SnIn_4S_8$ after long-term photocatalytic cycles have very little obvious change in the structure and morphology (Figures S4 and S5). In addition, the control experiments were carried out in the dark, in the Ar atmosphere and in the absence of photocatalysts (Figure 4c). There is nearly no product in the photocatalytic $CO_2$ reduction process, proving the fact that the carbon-containing products are actually triggered by the photocatalytic $CO_2$ reduction instead of adsorbed carbon species.

The photo absorption capacity of the as-obtained materials was investigated by the UV-vis diffuse reflectance spectrum (DRS). As shown in Figure 5a, $SnIn_4S_8$ displays a steep absorption region of about 650 nm in the visible light region owing to the narrow bandgap. After the loading of Au nanocrystals, the absorption edge of $Au_2$-$SnIn_4S_8$ does not appear as an extra absorption peak caused by the LSPR effect of the plasmonic metal. This can be illustrated by the fact that the dark yellow-green $SnIn_4S_8$ appears to have a strong absorption intensity and broad absorption range in the UV-vis region, which may cover up the plasmonic band of Au [16]. When the Cu and CuAu nanocrystals were introduced, the background absorption was significantly improved to the longer wavelength region, which greatly matches the color of the catalysts changing from dark yellow–green to black. Furthermore, the band gap (Eg) of the pure $SnIn_4S_8$ was calculated by a typical Tauc approach. As exhibited in Figure S6, the Eg of $SnIn_4S_8$ is estimated to be 1.92 eV, which is suitable for the process of a photocatalytic $CO_2$ reduction. In addition, the Mott–Schottky plots of $SnIn_4S_8$ were measured at 1500, 2000 and 2500 Hz, respectively, and the flat-band potential of $SnIn_4S_8$ is about –0.76 V (vs. NHE, pH = 7) (Figure S7). Generally speaking, the positive slope of the Mott–Schottky plot indicates that $SnIn_4S_8$ belongs to n-type semiconductors, whose flat-band position is nearly at the Fermi level [14,37]. As shown in green line in Figure S3, the gap between the Fermi level and the VB of $SnIn_4S_8$ is approximately 1.51 eV from the XPS valence band spectrum. Therefore, the VB value of $SnIn_4S_8$ was estimated to be 0.75 eV (vs. NHE, pH = 7). According to the following formula: $E_{CB} = E_{VB} - E_g$, the value of CB can be determined as –1.17 V (vs. NHE, pH = 7). In summary, the obtained band structure of photocatalyst conforms to the thermodynamic potential of a photocatalytic $CO_2$ reduction, and the photogenerated electrons in the CB of $SnIn_4S_8$ possess a negative potential enough for the photocatalytic $CO_2$ reduction.

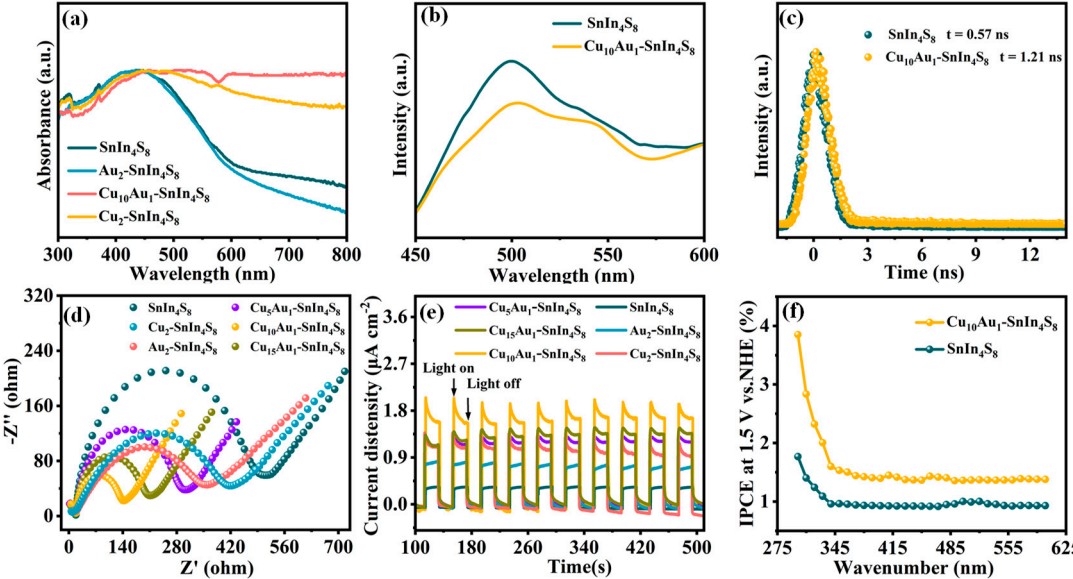

**Figure 5.** (**a**) UV-vis DRS spectra, (**b**) PL spectra, (**c**) TRPL spectra, (**d**) EIS and (**e**) transient photocurrent responses (**f**) IPCE of the photocatalysts.

Steady-state photoluminescence (PL) spectra are utilized to evaluate the photoexcited electrons and holes migration ability of the semiconductor photocatalyst. The decreasing PL emission intensity means a more effective inhibition of the electron-hole recombination [40]. Figure 5b shows the PL spectra of pure $SnIn_4S_8$ and $Cu_{10}Au_1$-$SnIn_4S_8$. The peak intensity of $Cu_{10}Au_1$-$SnIn_4S_8$ reduced remarkably as compared to $SnIn_4S_8$, suggesting that the loading of the CuAu alloy is able to effectively facilitate the separation and transfer of the electron-hole pairs. In addition, this enhanced charge transfer behavior was further confirmed by the time-resolved fluorescence emission decay spectra (TRPL). In Figure 5c, the average fluorescence lifetimes of $Cu_{10}Au_1$-$SnIn_4S_8$ (1.21 ns) are longer than that of pure $SnIn_4S_8$ (0.57 ns) because of the introduction of CuAu alloys. The extended carrier average lifetimes inevitably conduce to an accelerated separation and migration of photogenerated charge carriers, which is of great significance to promote the photoreduction $CO_2$ activity by offering more charge carriers to get involved in the photocatalytic process [41,42]. The above consequence was further confirmed by transient photocurrent test and electrochemical impedance spectroscopy (EIS). The different addition of metal nanocrystals on $SnIn_4S_8$ displayed a significant enhancement of the photocurrent intensity and decrease in the interfacial charge transfer resistances compared with the pure $SnIn_4S_8$, while $Cu_{10}Au_1$-$SnIn_4S_8$ possesses the smallest charge transfer resistance and the most obvious photocurrent density (Figure 5d,e). The electron trapping abilities are in the following order: $SnIn_4S_8$ < $Au_2$-$SnIn_4S_8$ < $Cu_2$-$SnIn_4S_8$ < $Cu_5Au_1$-$SnIn_4S_8$ < $Cu_{15}Au_1$-$SnIn_4S_8$ < $Cu_{10}Au_1$-$SnIn_4S_8$. In other words, the CuAu alloy cocatalyst is more beneficial to accelerate the photogenerated charge separation and prevent the recombination for a higher photocatalytic performance compared with the single metal cocatalyst [43]. At the same time, we carried out the incident photon-to-current conversion efficiency (IPCE) to reveal the photogenerated carrier separation and transfer efficiency (Figure 5f). Owing to the accelerated migration of photogenerated charge after Cu reacts with Au to produce the alloy, $Cu_{10}Au_1$-$SnIn_4S_8$ performs an improved IPCE from 300 to 600 nm, which greatly matches the photocatalytic activity of the samples [44].

According to the fact that the electron-hole separation was efficiently boosted significantly contributes to the photocatalytic performance, but as we all know, many factors are revealed to influence the $CO_2$ photoreduction activity. $CO_2$ capture and concentration by catalytic-active sites on the surface is another necessary factor. Hence, we used $N_2$ adsorption-desorption measurement to evaluate the specific surface areas of the materials. As revealed in Figure 6a, the surface areas of $SnIn_4S_8$, $Au_2$-$SnIn_4S_8$, $Cu_2$-$SnIn_4S_8$ and $Cu_{10}Au_1$-$SnIn_4S_8$ are, respectively, measured as 23.51, 16.35, 11.57 and 20.21 $m^2/g$, suggesting that the load of metal nanocrystals makes an effect on the decreased specific surface area. Additionally, we conducted $CO_2$ adsorption tests. As displayed in Figure 6b, the $Cu_{10}Au_1$-$SnIn_4S_8$ displays the maximum $CO_2$ adsorption performance, which is notably higher than $SnIn_4S_8$. Though the surface area of $Cu_{10}Au_1$-$SnIn_4S_8$ decreases, the $CO_2$ adsorption capability enhances, which can be due to the synergy effect between the CuAu alloy that improves the $CO_2$ adsorption ability [42]. Furthermore, linear sweep voltammetry (LSV) tests are carried out to confirm the effects of the loading of the alloy on the activation ability of the $CO_2$ molecules. From the results revealed in Figure 6c, $Cu_{10}Au_1$-$SnIn_4S_8$ enables a clearly reduced initial potential and relatively higher current density compared with pristine $SnIn_4S_8$ and monometallic nanoparticles integrated into the $SnIn_4S_8$, which shows that the cooperative effect of the Au and Cu can remarkably facilitate the activation of $CO_2$ molecules and the conversion of $CO_2$ into hydrocarbons [45]. The above results are coupled with the increased photocatalytic $CO_2$ reduction performance.

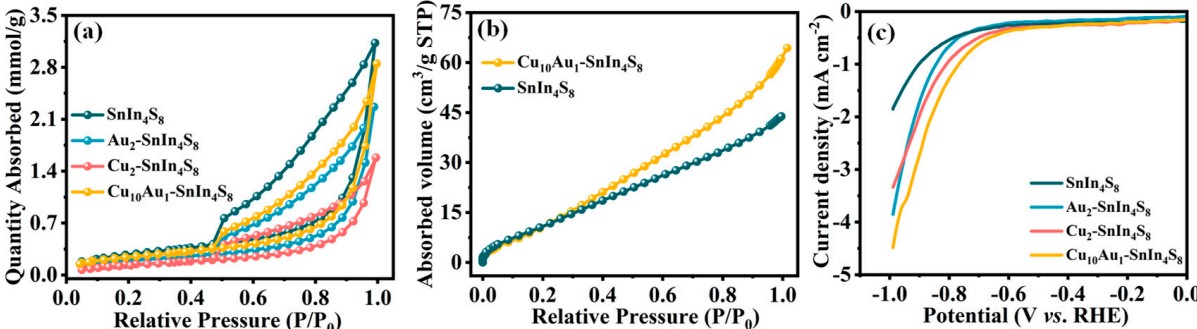

**Figure 6.** (**a**) $N_2$ adsorption–desorption isotherms of the samples, (**b**) $CO_2$ adsorption isotherms of $SnIn_4S_8$ and $Cu_{10}Au_1$-$SnIn_4S_8$, (**c**) LSV curves tested in $CO_2$ of the samples.

For the purpose of gaining deeper insight into the process of photocatalytic $CO_2$ reduction experiment, in situ Fourier transform infrared spectroscopy (in situ FTIR) experiments were utilized to investigate the key reaction intermediations. The in situ FTIR spectra of $SnIn_4S_8$ and $Cu_{10}Au_1$-$SnIn_4S_8$ were collected when filled with $CO_2$ gas in dark and under light irradiation for 10 min, 20 min and 30 min. By comparison (Figure 7a,b), the in situ FT-IR spectra of $SnIn_4S_8$ and $Cu_{10}Au_1$-$SnIn_4S_8$ both appear at different reaction intermediates, and two peaks located at 1213 and 1401 $cm^{-1}$ were assigned to the bicarbonate ($HCO_3^-$) species, and peaks centered on 1297 $cm^{-1}$ and 1353 $cm^{-1}$ were attributed to the b-$CO_3^{2-}$ and m-$CO_3^{2-}$. Additionally, the other two peaks detected at around 1448 $cm^{-1}$ and 1614 $cm^{-1}$ suggested the formation of key intermediate $CO_2^-$ groups [37,46]. All the peaks generated in the 1000–1800 $cm^{-1}$ range were enhanced progressively with the extension of the UV-vis irradiation time, implying that the $CO_2$ absorbed on the surface of the sample can interact with photoexcited electrons to produce $CO^{2-}$. Furthermore, $CO^{2-}$ is able to combine with $H^+$ to create *COOH, which is the key intermediation in the process of a $CO_2$ conversion into CO [47]. Especially, for $Cu_{10}Au_1$-$SnIn_4S_8$, the tendency of the peak intensities became obviously stronger, suggesting that the loading of the CuAu alloy can strengthen the ability of the $CO_2$ adsorption on the photocatalyst surface and facilitate the reduction efficiency of the photocatalytic $CO_2$ [42].

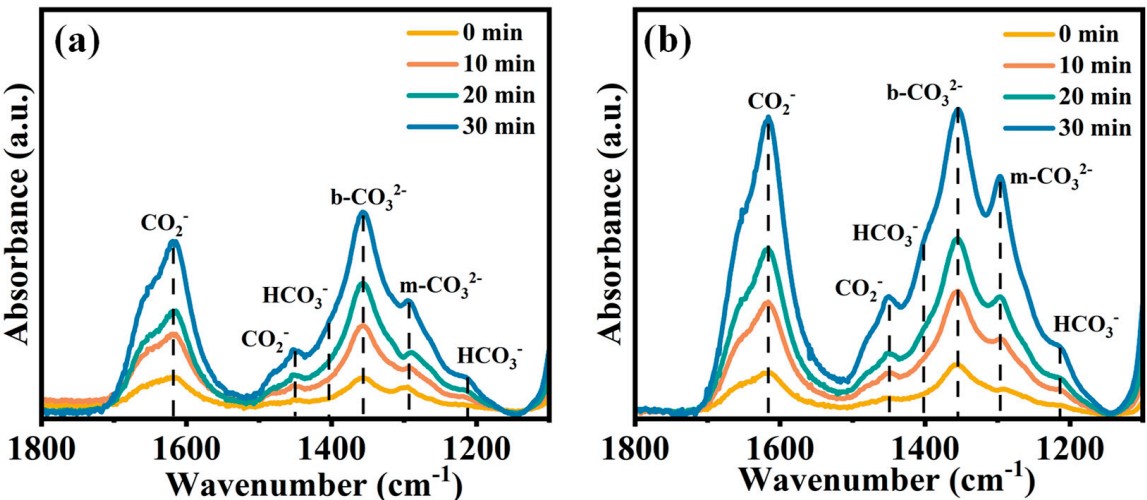

**Figure 7.** In situ FTIR spectra of (**a**) $SnIn_4S_8$ and (**b**) $Cu_{10}Au_1$-$SnIn_4S_8$.

The most possible $CO_2$ photoreduction pathways for $Cu_{10}Au_1$-$SnIn_4S_8$ can be speculated as follows:

$$* + CO_2 \rightarrow *CO_2 \tag{1}$$

$$CO_2* + e^- \rightarrow *CO_2^- \tag{2}$$

$$*CO_2^- + H^+ \rightarrow *COOH \tag{3}$$

$$*COOH + H^+ + e^- \rightarrow *CO + H_2O \tag{4}$$

$$*CO \rightarrow CO \tag{5}$$

$$*CO + H^+ + e^- \rightarrow *CHO \tag{6}$$

$$*CHO + 2H^+ + 2e^- \rightarrow *CH_3O \tag{7}$$

$$*CH_3O + 3H^+ + 3e^- \rightarrow *CH_4 + H_2O \tag{8}$$

$$*CH_4 \rightarrow CH_4 \tag{9}$$

Based on the above comprehensive experimental discussion, the possible mechanism for the photocatalytic $CO_2$ reduction on the CuAu-$SnIn_4S_8$ photocatalyst is presented in Figure 8. Under visible light irradiation, photoexcited electrons jump into the CB from the VB of $SnIn_4S_8$, causing many holes on the VB of $SnIn_4S_8$, and forming photoinduced electron-hole pairs. The photoexcited electrons then rapidly shift to the CuAu alloy nanoparticles due to their specific plasmonic property. Lastly, the electrons gather on the surface of CuAu alloys and then react with $CO_2$ to CO and $CH_4$. At the same time, the TEOA as the hole ($h^+$) sacrificial agent reacts with holes in the VB of $SnIn_4S_8$ to further restrain the recombination of photo-induced charge carriers.

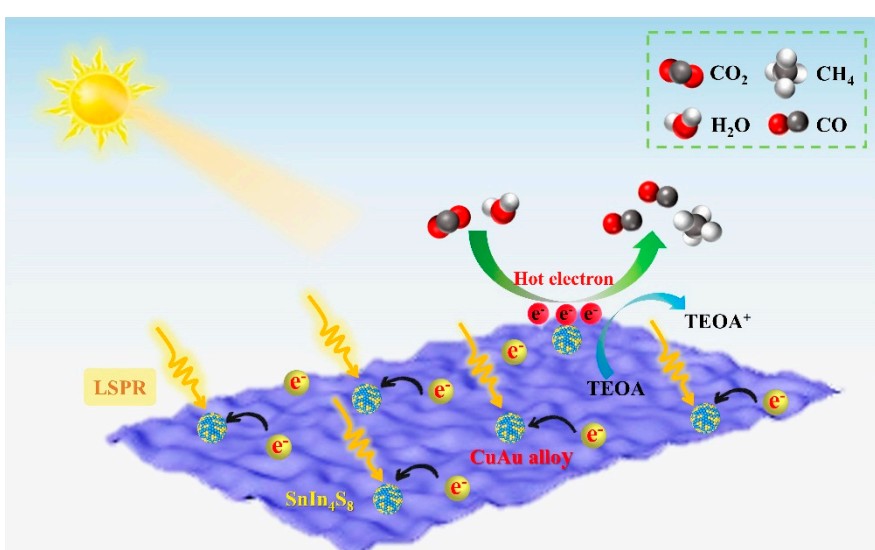

**Figure 8.** Schematic illustration of photocatalytic $CO_2$ reduction process on CuAu-$SnIn_4S_8$.

## 3. Conclusions

In summary, a novel CuAu alloy-loaded $SnIn_4S_8$ photocatalyst has been designed and fabricated through a facile aqueous solution method. The CuAu-$SnIn_4S_8$ photocatalyst displays a remarkably enhanced photocatalytic $CO_2$ reduction performance and stability as compared with pristine $SnIn_4S_8$ and single metal nanocrystals on $SnIn_4S_8$. The optimized performance originates from the presence of the CuAu alloy, which greatly improves the light absorption ability, acts as the synergistic site to reinforce the $CO_2$ adsorption and activation on the photocatalyst surface and effectively suppresses the recombination of the photoexcited charge. This work offers a useful ideal for the design of photocatalysts with

the synergistic function of the alloy cocatalysts for the high performance of a solar-driven $CO_2$ reduction.

**Supplementary Materials:** The following supporting information can be downloaded at: https://www.mdpi.com/article/10.3390/catal12121588/s1, Figure S1: Energy dispersive spectrometer of $Cu_{10}Au_1$-$SnIn_4S_8$; Figure S2: Survey spectra of $SnIn_4S_8$ and $Cu_{10}Au_1$-$SnIn_4S_8$; Figure S3: XPS valence band spectra of $SnIn_4S_8$ and $Cu_{10}Au_1$-$SnIn_4S_8$; Figure S4: XRD patterns of $Cu_{10}Au_1$-$SnIn_4S_8$ before and after photocatalytic $CO_2$ reduction reaction; Figure S5: SEM image of $Cu_{10}Au_1$-$SnIn_4S_8$ after photocatalytic $CO_2$ reduction reaction; Figure S6: Tauc plots of $SnIn_4S_8$; Figure S7: Mott-Schottky plots of $SnIn_4S_8$; Table S1. Comparison of the $CO_2$ photoreduction performance of $Cu_{10}Au_1$-$SnIn_4S_8$ catalyst with other catalysts [22,23,41,48–52].

**Author Contributions:** Z.Y.: writing—original draft. J.Y.: writing—review and editing. K.Y.: investigation. X.Z.: visualization. K.Z.: visualization. M.Z.: validation and formal analysis. H.J.: resources. M.H.: resources. H.L.: resources. H.X.: conceptualization, methodology, and funding acquisition. All authors have read and agreed to the published version of the manuscript.

**Funding:** This research was funded by the Natural Science Foundation of Jiangsu Province (BK20221367), National Natural Science Foundation of China (22075113, 22138011), High-tech Research Key Laboratory of Zhenjiang (SS2018002), Jiangsu Provincial Agricultural Science and Technology Independent Innovation Fund (CX(21)3067), Natural Science Foundation of Jiangsu Province (BK20220598).

**Data Availability Statement:** Data is contained within the article and Supplementary Materials.

**Conflicts of Interest:** The authors declare no conflict of interest.

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
