# Peer review of "Synergistic Effect in Plasmonic CuAu Alloys as Co-Catalyst on SnIn4S8 for Boosted Solar-Driven CO2 Reduction"

_catalysts, doi:10.3390/catal12121588_

Round 1
Reviewer 1 Report
The article is devoted to the study of the SnIn4S8 composite photocatalyst decorated with CuAu alloy nanoparticles. An increase in the efficiency of the SnIn4S8 semiconductor catalyst due to the plasmonic properties of CuAu alloy nanoparticles and charge separation by the metal/semiconductor interface is shown. The article demonstrates detailed studies of the obtained catalytic material and interesting new results.
In general, with a good preparation of the article, there is one fundamental remark: the article does not contain the section "Materials and Methods" with a description of the details of the composite catalytic material formation. In addition, the designations Cu10Au1, Au2, Cu2 require clarification.
There is a typo in line 86: C10Au1-SnIn4S8.
Author Response
Dear Editor and Reviewers:
Thank you very much for your letter and the comments from the Reviewers about our paper submitted to Catalysts (ID: catalysts-2054916, Title: “Synergistic effect in plasmonic CuAu alloys as co-catalyst on SnIn4S8 for boosted solar-driven CO2 reduction”). In order to highlight the changes that we have made in the paper, the color of the text changed in the paper has become blue.
We would like to thank the reviewer for the careful reading of our manuscript. We have revised the paper according to the comments of the Reviewers. We submit the revised manuscript here as well as a list of changes. Our responses to Reviewers’ comments are listed as follows.
Answer: Thanks for your nice suggestion. We have provided the section of Materials and Methods in the supporting information. In the supporting information, we have made a detailed explanation about the clarification of the designations Cu10Au1, Au2, Cu2. The total amount of CuCl2·2H2O solution (0.1mol/L) and HAuCl4·4H2O solution (50 mmol/L) is 0.2 mmol. For comparison, a sequence of CuxAu1-SnIn4S8 samples was obtained, which were labeled as Au2-SIS, Cu5Au1-SIS, Cu10Au1-SIS, Cu15Au1-SIS and Cu2-SIS, where x and y corresponding to the molar ratio of CuCl2·2H2O and HAuCl4·4H2O, respectively.
The revision is as follows:
Among the designed samples, the Cu10Au1-SnIn4S8 achieved the activity of 27.87 μmol g-1 h-1 of CO and 7.21 μmol g-1 h-1 of CH4, which are about 7.6 and 2.5 folds compared with SnIn4S8.

Reviewer 2 Report
1. In the introduction part, it is better to give the reasons why SnIn4S8 was chosen as support in this study.
2. How will the composition of CuAu alloy influence the performance of the catalysts? This study focused on Cu10Au1 only.
3. As the size of plasmonic nanoparticles and the distance between two nanoparticles influence the electromagnetic field of the catalysts, it is better to provide these informations.
4. How was the performance without TEOA? Will the reaction occur with CO2+H2O only?
5. More relevant reference should be cited. CATALYSIS SCIENCE & TECHNOLOGY, 2022, 12, 6155-6162; ANGEWANDTE CHEMIE-INTERNATIONAL EDITION, 2015, 54, 11545-11549.
Author Response
Dear Editor and Reviewers:
Thank you very much for your letter and the comments from the Reviewers about our paper submitted to Catalysts (ID: catalysts-2054916, Title: “Synergistic effect in plasmonic CuAu alloys as co-catalyst on SnIn4S8 for boosted solar-driven CO2 reduction”). In order to highlight the changes that we have made in the paper, the color of the text changed in the paper has become blue.
We would like to thank the reviewer for the careful reading of our manuscript. We have revised the paper according to the comments of the Reviewers. We submit the revised manuscript here as well as a list of changes. Our responses to Reviewers’ comments are listed as follows.
Q1: In the introduction part, it is better to give the reasons why SnIn4S8 was chosen as support in this study.
Answer: Thanks for your excellent suggestion. For this question, we have made a detailed explanation in the introduction part.
Among various reported semiconductors photocatalysts previously, bimetallic sulfides have aroused great scientific interest due to their unique characteristics, for instance, adjustable morphology, enriched active sites, controllable band structure and fast photoexcited charge dynamics [7, 8]. Particularly, SnIn4S8 is a typical n-type bimetallic sulfide semiconductor with a cubic spinel structure [9]. Featured by the narrow band gap, easily adjustable electronic and optical properties, SnIn4S8 is a suitable candidate in high-energy batteries and photocatalytic applications [10, 11]. The above advantages inspire us to consider their potential roles in photocatalytic CO2 reduction. In addition, the poor light utilization and short lifetime of photoinduced carriers seriously influence the photocatalytic ability in visible light. Therefore, SnIn4S8 requires to be modified to suppress fast recombination of photoinduced carriers and enhance the ability to absorb and activate CO2 molecules [12].
Q2: How will the composition of CuAu alloy influence the performance of the catalysts? This study focused on Cu10Au1 only.
Answer: Thanks for your nice suggestion. We are genuinely sorry for making such mistakes due to oversight. In the activity test, we tested the activity of various amount of CuAu alloy loaded SnIn4S8. Obviously, the different composition of CuAu alloys was beneficial to the photoreduction process. Cu10Au1-SnIn4S8 was chosen as the representative because it showed the best photocatalytic performance among Cu5Au1-SnIn4S8, Cu10Au1-SnIn4S8 and Cu15Au1-SnIn4S8. Based on the reviewer's comments, we have also carried out the EIS and transient photocurrent responses of Cu5Au1-SnIn4S8 and Cu15Au1-SnIn4S8 photocatalysts to study the electron trapping abilities (Figure 5), which are in the following order: SnIn4S8 < Au2-SnIn4S8 <Cu2-SnIn4S8 < Cu5Au1-SnIn4S8 < Cu15Au1-SnIn4S8 < Cu10Au1-SnIn4S8. It demonstrates that the various CuAu alloys significantly enhances the efficiency of electron-hole separation. and the different charge separation behaviors can be used to explain the improved CH4 production of Cu10Au1-SnIn4S8. Cu has been previously identified as a metal which can provide more elections for the conversion of CO2 to CH4, thus the higher content of Cu contributes to the stronger electron trapping ability of cocatalysts. The conversion of CO2 to CH4 (CO2 +8H+ +8e- / CH4 +2H2O) is an 8-electron process, so the enhanced electron trapping can make an important contribution. This work attaches importance to the loading of CuAu alloys as co-catalyst for boosted photocatalytic CO2 reduction. At present, we are also making relevant research about the composition of CuAu alloy influence the performance, and hope to have the opportunity to present meaningful results to you for your review in the future.
Q3: As the size of plasmonic nanoparticles and the distance between two nanoparticles influence the electromagnetic field of the catalysts, it is better to provide these information.
Answer: Thanks for your nice suggestion. From the TEM images in Figure 2b, Cu10Au1 alloy nanoparticles show a round-shaped form with a uniform particle diameter of about 8 nm. Figure 2l presents the HRTEM image of a representative CuAu alloy, the CuAu alloy nanoparticle shows the lattice distances of 0.24 and 0.21 nm, respectively corresponding with the (201) and (211) planes for CuAu alloys, indicating that the CuAu alloys are successfully loaded on the surface of SnIn4S8. (J. Am. Chem. Soc. 2014, 136, 15969-15976; ACS Nano, 9, 14453-14464). The SPR effect of the metal is related to its particle size, and the SPR effect of the metal increases with the increased particle size in a certain size range. The Cu10Au1- SnIn4S8 photocatalyst shows strong absorption in the visible light region, indicating that CuAu alloy NPs possess a strong SPR effect.
Q4: How was the performance without TEOA? Will the reaction occur with CO2+H2O only?
Answer: Thanks for your suggestions. In this photocatalysis system, TEOA is used as the hole sacrificial agent, which is also named the electron donor. The function of TEOA is to react with holes inhibiting electron-hole pair recombination. Due to the defect of easy recombination of electron-hole pairs in the bulk phase of bimetallic sulfide materials, TEOA is indispensable in most photocatalytic CO2 reduction systems with bimetallic sulfide as the photocatalyst. (Angew. Chem. Int. Ed. 2022, 61, e2022107; Chemical Engineering Journal, 2021, 418, 129476) When no hole sacrificial agent is involved, the catalytic performance is very low. (J. Mater. Chem. A, 2020, 36, 18707-18714; J. Mater. Chem. A, 2020, 15, 7177-7183; J. Am. Chem. Soc., 2018, 140, 5037-5040).
Thus, more efforts should be put into inhibiting the photogenerated carrier recombination of carbon nitride materials to improve the photocatalytic CO2 reduction performance in pure water systems. At present, we are also doing relevant research, and hope to have the opportunity to present meaningful results to you for your review in the future.
Q5: More relevant reference should be cited. CATALYSIS SCIENCE & TECHNOLOGY, 2022, 12, 6155-6162; ANGEWANDTE CHEMIE-INTERNATIONAL EDITION, 2015, 54, 11545-11549.
Answer: Thanks for your valuable suggestion. Many thanks for the recommendation of the recent report, which could make the manuscript more convincing. We have cited some recent report and added new ref. 30 and ref. 31 in the main text and more in the supporting information.
[30] L.Z. Shi, H.M. Liu, S.B. Ning, J.H. Ye, Localized surface plasmon resonance effect enhanced Cu/TiO2 core-shell catalyst for boosting CO2 hydrogenation reaction, Catal. Sci. Technol., 2022,12, 6155-6162.
[31] H. Liu, X. Meng, T.D. Dao, H. Zhang, P. Li, K. Chang, T. Wang, M. Li, T. Nagao, J. Ye, Conversion of Carbon Dioxide by Methane Reforming under Visible-Light Irradiation: Surface-Plasmon-Mediated Nonpolar Molecule Activation, Angew Chem Int. Ed. Engl., 2015, 54, 11545-11549.

Reviewer 3 Report
This manuscript reports the synthesis, characterization and photocatalytic CO2 reduction activity of CuAu-SnIn4S8. The introduction CuAu alloys extends the light absorption and promotes the adsorption and activation of CO2 and efficient separation and migration of carriers. In addition, it is instructive to systematically study the structure and composition of CuAu-SnIn4S8 in this work. Attractively, CuAu-SnIn4S8 manifested significantly improved photocatalytic CO2 reduction activity under 300 W Xe lamps. The manuscript is interesting and in-depth. I recommend it for acceptance for publication in Catalysts after the following revisions.
1. In Figure 3, it is necessary to state the full name of catalysts. Check it carefully and amend it.
2. The recent advances regarding alloy as co-catalyst should be cited in the section of the introduction to highlight the innovation.
3. The obtained activities should be compared with the recently published data on other types of photocatalysts, for example, indium-based ternary metal sulfides.
4. The transient photocurrent responses spectra in Figure 5e should provide the sign of light on and light off.
5. What role does TEOA play in CO2 photoreduction and the mechanism of the best performance of CuAu-SnIn4S8 needs to be further refined and optimized.
Author Response
Dear Editor and Reviewers:
Thank you very much for your letter and the comments from the Reviewers about our paper submitted to Catalysts (ID: catalysts-2054916, Title: “Synergistic effect in plasmonic CuAu alloys as co-catalyst on SnIn4S8 for boosted solar-driven CO2 reduction”). In order to highlight the changes that we have made in the paper, the color of the text changed in the paper has become blue.
We would like to thank the reviewer for the careful reading of our manuscript. We have revised the paper according to the comments of the Reviewers. We submit the revised manuscript here as well as a list of changes. Our responses to Reviewers’ comments are listed as follows.
Q1: In Figure 3, it is necessary to state the full name of catalysts. Check it carefully and amend it.
Answer: Thanks very much for your important suggestions. According to your suggestion, we have added the full name in Figure 3.
Q2: The recent advances regarding alloy as co-catalyst should be cited in the section of the introduction to highlight the innovation.
Answer: Thanks for your valuable suggestion. Many thanks for the recommendation of the recent report, which could make the introduction more innovative. We have cited some recent report and added new ref. 18, ref. 19, ref. 20 in the main text.
[18] R.S. Haider, S. Wang, Y. Gao, A.S. Malik, N. Ta, H. Li, B. Zeng, M. Dupuis, F. Fan, C. Li, Boosting photocatalytic water oxidation by surface plasmon resonance of AgxAu1−x alloy nanoparticles, Nano Energy, 87 (2021).
[19] S. Fu, C. Zhu, Q. Shi, H. Xia, D. Du, Y. Lin, Highly branched PtCu bimetallic alloy nanodendrites with superior electrocatalytic activities for oxygen reduction reactions, NANOSCALE, 8 (2016) 5076-5081.
[20] S. Lee, S. Jeong, W.D. Kim, S. Lee, K. Lee, W.K. Bae, J.H. Moon, S. Lee, D.C. Lee, Low-coordinated surface atoms of CuPt alloy cocatalysts on TiO2 for enhanced photocatalytic conversion of CO2, NANOSCALE, 8 (2016) 10043-10048.
Q3: The obtained activities should be compared with the recently published data on other types of photocatalysts, for example, indium-based ternary metal sulfides.
Answer: Thanks very much for your nice suggestions. We have added a brief discussion about the obtained activities compared with the recently published data on other types of photocatalysts.
The revision is as follows:
Besides, the latest photocatalysts for CO2 reduction and their abilities are listed in Table S1. The performance of Cu10Au1-SnIn4S8 is in a competitive position.
Table S1. Comparison of the CO2 photoreduction performance of Cu10Au1-SnIn4S8 catalyst with other catalysts.
Photocatalyst |
light source |
Experimental conductions |
Activity (μmol·g-1·h-1) |
Reference |
10mg of Cu10Au1-SnIn4S8 |
300 W Xe lamp |
H2O, TEOA |
CO: 27.87 |
This work |
10 mg of In2S3-CuInS2 |
300 W Xe lamp |
CoCl2, 2,2-bipyridine, TEOA, MeCN |
CO: 19 |
[1] |
10mg of WQDs/CdIn2S4 |
300 W Xe lamp |
water vapor |
CO: 8.2 CH4: 1.6 |
[2] |
30 mg of In2O3/In2S3 |
300 W Xe lamp |
H2O |
CO: 12.22 |
[3] |
15 mg of TiO2-AuCu-V |
300 W Xe lamp |
H2O |
CH4: 33.5 |
[4] |
5mg of CN-PA12 |
AM 1.5 illumination |
water vapor |
CO: 5.42 CH4: 4.03 |
[5] |
0.1g ZnIn2S4/BiVO4 |
300 W Xe lamp |
water vapor |
CO: 4.75, CH4: 0.5 |
[6] |
0.1g of ZnIn2S4/ N-doped graphene |
300 W Xe lamp |
water |
CO: 2.45 CH4: 1.01 CH3OH: 1.37 |
[7] |
50mg of CuInS2/Au/g-C3N4 |
300 W Xe lamp 400 nm cutoff filter |
water vapor |
CO: 2.43 CH4: 0.15 |
[8] |
References
[1] J. Yang, X. Zhu, Z. Mo, J. Yi, J. Yan, J. Deng, Y. Xu, Y. She, J. Qian, H. Xu, H. Li, A multidimensional In2S3-CuInS2 heterostructure for photocatalytic carbon dioxide reduction, Inorg. Chem. Front., 5 (2018) 3163-3169.
[2] Z. Zhang, Y. Cao, F. Zhang, W. Li, Y. Li, H. Yu, M. Wang, H. Yu, Tungsten oxide quantum dots deposited onto ultrathin CdIn2S4 nanosheets for efficient S-scheme photocatalytic CO2 reduction via cascade charge transfer, Chem. Eng. J., 428 (2022) 131218.
[3] J. Yang, X. Zhu, Q. Yu, M. He, W. Zhang, Z. Mo, J. Yuan, Y. She, H. Xu, H. Li, Multidimensional In2O3/In2S3 heterojunction with lattice distortion for CO2 photoconversion, Chinese J. Catal., 43 (2022) 1286-1294.
[4] Q. Liu, Q. Chen, T. Li, Q. Ren, S. Zhong, Y. Zhao, S. Bai, Vacancy engineering of AuCu cocatalysts for improving the photocatalytic conversion of CO2 to CH4, J. Mater. Chem. A, 7 (2019) 27007-27015.
[5] Z. Wang, H. Lee, J. Chen, M. Wu, D.Y.C. Leung, C.A. Grimes, Z. Lu, Z. Xu, S.-P. Feng, Synergistic effects of Pd-Ag bimetals and g-C3N4 photocatalysts for selective and efficient conversion of gaseous CO2, J. Power Sources, 466 (2020) 228306.
[6] Q. Han, L. Li, W. Gao, Y. Shen, L. Wang, Y. Zhang, X. Wang, Q. Shen, Y. Xiong, Y. Zhou, Z. Zou, Elegant Construction of ZnIn2S4/BiVO4 Hierarchical Heterostructures as Direct Z-Scheme Photocatalysts for Efficient CO2 Photoreduction, ACS Appl. Mater. Interfaces, 13 (2021) 15092-15100.
[7] Y. Xia, B. Cheng, J. Fan, J. Yu, G. Liu, Near-infrared absorbing 2D/3D ZnIn2S4/N-doped graphene photocatalyst for highly efficient CO2 capture and photocatalytic reduction, Sci. China Mater., 63 (2020) 552-565.
[8] W. Ye, J. Hu, X. Hu, W. Zhang, X. Ma, H. Wang, Rational Construction of Z-Scheme CuInS2/Au/g-C3N4 Heterostructure: Experimental Results and Theoretical Calculation, ChemCatChem, 11 (2019) 6372-6383.
Q4: The transient photocurrent responses spectra in Figure 5e should provide the sign of light on and light off.
Answer: Thanks very much for your suggestions. We have added it in Figure 5e.
Figure 5 (a) UV-vis DRS spectra, (b) PL spectra, (c) TRPL spectra, (d) EIS and (e) transient photocurrent responses (f) IPCE of the photocatalysts.
Q5: What role does TEOA play in the CO2 photoreduction and the mechanism of the best performance of CuAu-SnIn4S8 needs to be further refined and optimized.
Answer: Thanks for your suggestions. In the system of CO2 reduction, TEOA is used as the hole sacrificial agent. (J. Am. Chem. Soc. 2013, 135, 5441-5449; J. Am. Chem. Soc. 2015, 137, 13440-13443; J. Am. Chem. Soc. 2015, 137, 604−607; J. Am. Chem. Soc. 2017, 139, 17305-17308; J. Am. Chem. Soc. 2018, 140, 5037-5040).
In the part of the discussion of photocatalytic mechanism, we have added a detailed explanation:
Based on the above comprehensive experimental discussion, the possible mechanism for photocatalytic CO2 reduction on CuAu-SnIn4S8 photocatalyst is presented in Figure 9. Under visible light irradiation, photoexcited electrons jump into the CB from the VB of SnIn4S8, causing many holes on the VB of SnIn4S8, and forming photoinduced electron-hole pairs. The photoexcited electrons then rapidly shift to CuAu alloy nanoparticles due to their specific plasmonic property. Lastly, the electrons gather on the surface of CuAu alloys and then react with CO2 to CO and CH4. At the same time, the TEOA as the hole (h+) sacrificial agent reacts with holes in the VB of SnIn4S8 to further restrain the recombination of photo-induced charge carriers.
